# Unveiling the Genetic Landscape of Canine Papillomavirus in the Brazilian Amazon

**DOI:** 10.3390/microorganisms13081811

**Published:** 2025-08-02

**Authors:** Jeneffer Caroline de Macêdo Sousa, André de Medeiros Costa Lins, Fernanda dos Anjos Souza, Higor Ortiz Manoel, Cleyton Silva de Araújo, Lorena Yanet Cáceres Tomaya, Paulo Henrique Gilio Gasparotto, Vyctoria Malayhka de Abreu Góes Pereira, Acácio Duarte Pacheco, Fernando Rosado Spilki, Mariana Soares da Silva, Felipe Masiero Salvarani, Cláudio Wageck Canal, Flavio Roberto Chaves da Silva, Cíntia Daudt

**Affiliations:** 1Laboratório de Virologia Geral e Parasitologia, Centro de Ciências Biológicas e da Natureza, Universidade Federal do Acre, Campus Universitário, BR 364, Km 04-Distrito Industrial, Rio Branco 69920-900, AC, Brazil; jeneffercmacedo@gmail.com (J.C.d.M.S.); fernanda.anjos.asc@gmail.com (F.d.A.S.); higormanoel@hotmail.com (H.O.M.); cleytonsilvaaraujo92@gmail.com (C.S.d.A.); paulogiliogasparotto@gmail.com (P.H.G.G.); acacio.pacheco@ufac.br (A.D.P.); flavio.silva@ufac.br (F.R.C.d.S.); 2Laboratório de Medicina Veterinária Preventiva e Vacinologia, Instituto de Medicina Veterinária, Universidade Federal do Pará, Castanhal 68740-970, PA, Brazil; andre.lins@castanhal.ufpa.br; 3Centro de Ciências Exatas e Tecnológicas, Universidade Federal do Acre, Campus Universitário, BR 364, Km 04-Distrito Industrial, Rio Branco 69920-900, AC, Brazil; lorena.tomaya@ufac.br; 4Laboratório de Microbiologia Molecular, Universidade Feevale, Novo Hamburgo 93525-075, RS, Brazil; vyctoriamalayhkaa@gmail.com (V.M.d.A.G.P.); fernandors@feevale.br (F.R.S.); marianasilva2@feevale.br (M.S.d.S.); 5Laboratório de Virologia, Faculdade de Veterinária, Universidade Federal do Rio Grande do Sul, Av. Bento Gonçalves, 9090, Prédio 42.505, Porto Alegre 91540-000, RS, Brazil; claudio.canal@ufrgs.br

**Keywords:** canine virology, viral genomics, CPV1 diversity, molecular diagnostics, histopathological findings, papillomatous lesions, Amazon biodiversity, phylogenetic profiling, cutaneous papillomatosis, oral wart characterization

## Abstract

Papillomaviruses (PVs) are double-stranded DNA viruses known to induce a variety of epithelial lesions in dogs, ranging from benign hyperplasia to malignancies. In regions of rich biodiversity such as the Western Amazon, data on the circulation and genetic composition of canine papillomaviruses (CPVs) remain scarce. This study investigated CPV types present in oral and cutaneous papillomatous lesions in domiciled dogs from Acre and Rondônia States, Brazil. Sixty-one dogs with macroscopically consistent lesions were clinically evaluated, and tissue samples were collected for histopathological examination and PCR targeting the *L1* gene. Among these, 37% were histologically diagnosed as squamous papillomas or fibropapillomas, and 49.2% (30/61) tested positive for papillomavirus DNA. Sequencing of the *L1* gene revealed that most positive samples belonged to CPV1 (*Lambdapapillomavirus* 2), while one case was identified as CPV8 (*Chipapillomavirus* 3). Complete genomes of three CPV1 strains were obtained via high-throughput sequencing and showed high identity with CPV1 strains from other Brazilian regions. Phylogenetic analysis confirmed close genetic relationships among isolates across distinct geographic areas. These findings demonstrate the circulation of genetically conserved CPVs in the Amazon and reinforce the value of molecular and histopathological approaches for the accurate diagnosis and surveillance of viral diseases in domestic dogs, especially in ecologically complex regions.

## 1. Introduction

Papillomaviruses (PVs) are small, circular, double-stranded DNA viruses known to cause both pre-neoplastic and neoplastic diseases across various species, including humans, cats, horses, and dogs [1,2]. Classified within the *Papillomaviridae* family, PVs are categorized based on the nucleotide sequence similarity of the *L1* gene, biological properties, and phylogenetic tree topology [1,2]. In dogs, most infections with *Canis familiaris Papillomavirus* (canine papillomavirus—CPV) are asymptomatic [3]. However, CPVs have been implicated in causing conditions such as hyperplastic lesions [4] and, less frequently, neoplastic diseases [5]. Currently, 33 CPV types have been identified and are divided into three genera—*Lambdapapillomavirus*, *Taupapillomavirus*, and *Chipapillomavirus*—according to the Papillomavirus Episteme (PaVE) database (https://pave.niaid.nih.gov, accessed on 2 June 2025), with some types still unclassified [6].

CPVs are associated with various lesions, including oral and cutaneous papillomatosis, pigmented plaques, and squamous cell carcinomas (SCCs) [7,8,9,10,11]. While most canine pigmented plaques are small and localized, cases of extensive and disseminated plaques have been reported, with rare instances progressing to SCCs [7]. Specific CPV types, such as CPV16, are believed to have a higher potential for neoplastic transformations, although other types have also been found in lesions undergoing malignant changes [3,7,10,11,12]. In contrast, according to PaVE, over 400 human papillomavirus (HPV) types have been characterized. Despite recent reports of new viral types and co-infections, CPV diversity remains notably low. Factors such as viral evolution, anthropogenic landscape changes, and global climate shifts significantly influence the incidence and geographic distribution of viral agents affecting both animal and human populations.

The Amazon region, with its unparalleled biodiversity and complex ecosystems, highlights the critical need for virology research. Understanding how environmental factors contribute to the emergence and transmission of various viral pathogens is essential. Therefore, there is an urgent need for studies analyzing the genetic diversity of CPVs, particularly in the Amazon, which may influence virus–host dynamics in ways not observed in more urbanized or temperate areas, and contribute to a better understanding of CPV diversity, evolution, and disease expression. Thus, this study aims to identify the papillomavirus types present in oral and cutaneous papillomatous lesions in dogs from the Western Amazon region of Brazil. By expanding our knowledge of CPV diversity and its potential health implications, this research will provide valuable insights into canine virology and inform veterinary practices in biodiversity-rich areas.

## 2. Materials and Methods

### 2.1. Clinical Sampling and Histopathological Evaluation

This study included 61 domiciled dogs clinically presenting with papillomatous lesions—38 with cutaneous and 23 with oral lesions—originating from Acre (*n* = 18) and Rondônia (*n* = 43) states in the Western Brazilian Amazon. The dogs were voluntarily brought by their owners to private and university-affiliated veterinary clinics in both states for routine clinical evaluation. During consultation, licensed veterinarians identified lesions with exophytic, verrucous, or nodular morphologies indicative of papillomatosis. Upon clinical suspicion, and with informed consent from the owners, the animals were referred for further diagnostic investigation.

To ensure a minimally invasive procedure, lesion samples were collected under local infiltrative anesthesia using 2% lidocaine without a vasoconstrictor (Ceva, São Paulo, SP, Brazil). Sterile scalpels and forceps were used for excision. Following collection, each lesion was divided: one fragment was stored at –20 °C for subsequent DNA extraction, while 54 lesion samples, depending on tissue availability, were fixed in 10% buffered formaldehyde for histopathological examination.

Detailed clinical data were recorded for all dogs, including sex, age, breed, anatomical location, and the macroscopic morphology of lesions. Based on visual examination, lesions were classified into six distinct types: (1) cauliflower-like—irregular with a broad base; (2) filiform—thin projections resembling grains of rice; (3) nodular—raised, 1–2 cm in diameter; (4) dome-shaped—small, endophytic, approximately 4 mm; (5) papular—small (≈2 mm), flat lesions; and (6) pigmented plaques—hyperpigmented patches of skin.

To minimize animal suffering, all procedures were carried out in accordance with the European Convention for the Protection of Vertebrate Animals Used for Experimental and Other Scientific Purposes (2010/63/EU, revised 2010) and in accordance with the Colégio Brasileiro de Experimentação Animal (COBEA). The project was approved by the Comissão de Ética no Uso de Animais da Universidade Federal do Acre (protocol number 23107.005499/2018-96).

For histopathological processing, formalin-fixed tissues were kept in solution for at least 72 h, processed routinely, and sectioned at a 3 μm thickness. The sections were stained with hematoxylin and eosin (H&E) and examined microscopically by a veterinary pathologist. A visual overview of the clinical sampling and histopathological workflow is presented in Figure 1.

### 2.2. Molecular Analyses: DNA Extraction, Sequencing, and Phylogenetic Inference

Approximately 25 mg of each lesion was used for DNA extraction using the PureLink^®^ Genomic DNA Mini Kit (Invitrogen, Carlsbad, CA, USA), following the manufacturer’s protocol. Extracted DNA was stored at –20 °C.

To detect papillomaviral DNA, partial amplification of the *L1* gene was performed using degenerate primers FAP59 and FAP64, as previously described [13]. PCR products were purified using the PureLink^®^ Quick PCR Purification Kit (Invitrogen, Carlsbad, CA, USA) and sequenced using the ABI PRISM 3100 Genetic Analyzer (Applied Biosystems, Foster City, CA, USA) with the BigDye Terminator v3.1 Cycle Sequencing Kit. The resulting data were processed with Data Collection 3 software and converted to the FASTA format using Sequence Analysis Software v6 (Applied Biosystems, Foster City, CA, USA). Consensus sequences were assembled using Geneious Prime software (version 2025.2), and sequence identity was assessed via BLASTn and BLASTx against GenBank databases.

To obtain the complete viral genomes, rolling circle amplification (RCA) was applied to three samples with higher DNA quality and concentration using the TempliPhi™100 Amplification Kit (GE Healthcare, Burlington, MA, USA) [14,15]. The RCA products were purified, and their concentration and purity were measured using NanoDrop™ and Qubit™ systems (Invitrogen, Carlsbad, CA, USA). DNA libraries were prepared using 50 ng of purified DNA with the Nextera XT Sample Preparation Kit (Illumina, San Diego, CA, USA) and sequenced on an Illumina MiSeq platform using the v2 reagent kit (300 cycles).

The sequence data quality was evaluated with FastQC. Reads were trimmed at the 3′ end using a Phred quality threshold of 20 and assembled into contigs with SPAdes v3.5. Chimeric sequences were excluded based on RDP4 (version 4.1) software analysis [16]. Assembled sequences were annotated and analyzed using Geneious Prime (version 2025.2), and alignment with known papillomavirus sequences was performed using MAFFT (version 7) [17]. The final phylogenetic trees were constructed using the maximum likelihood method with the HKY model and 1000 bootstrap replicates in IQ-TREE [18].

### 2.3. Statistical Analysis

Given the heterogeneity of some categorical subgroups, preliminary data inspection guided the grouping of categories to ensure statistical robustness and meaningful comparisons. Specifically, the variable breed was consolidated into three categories: (1) mixed breeds, (2) small popular breeds (e.g., Poodle, Pinscher), and (3) medium/large popular breeds (e.g., Labrador, Pitbull). Similarly, the anatomical location of lesions was grouped into three regions: head (including ear, eye, and oral cavity), limbs, and torso (including neck, chest, back, abdomen, and vulva). For age, dogs were categorized as ≤3 years (juvenile/young adults) or >3 years (adults/seniors), based on biological relevance and sample distribution.

Statistical associations between papillomavirus detection (PCR positive or negative) and clinical or pathological variables were evaluated using Fisher’s Exact Test for 2 × 2 contingency tables (e.g., sex, age group, state, and histopathological diagnosis), as recommended for small samples and sparse data [19]. For variables with more than two categories—including breed group, anatomical location, and lesion morphology—Monte Carlo simulation with 10,000 replications was applied to estimate *p*-values [20]. All analyses were performed using R software version 4.5.1 [21], and differences were considered statistically significant when *p* < 0.05.

## 3. Results

### 3.1. Overview of Clinical and Histopathological Findings

Among the 61 dogs evaluated, clinically evident papillomatous lesions were primarily distributed in the oral cavity and limbs. The lesions varied in morphology, including cauliflower-like, nodular, filiform, and pigmented plaque presentations. Table 1 compiles all individual case data, including PCR results (including phylogeny findings), breed, age, sex, lesion location, and detailed macroscopic and histological characterizations.

Histopathological analysis was performed on 54 samples (88.5%), of which 20 (37%) were diagnosed as squamous papilloma (SP). These cases exhibited hallmark features such as epidermal hyperplasia, marked hyperkeratosis, and hypergranulosis (Figure 2a). In some instances, there was fibroblast proliferation within the dermis (Figure 2b), as well as lymphoplasmacytic inflammatory infiltrates, keratohyalin granules, and intranuclear inclusions—consistent with papillomavirus etiology.

### 3.2. Molecular Detection and Epidemiological Correlates

Of the 61 samples, 30 (49.2%) tested positive for papillomavirus DNA using PCR, including both cutaneous and oral lesions. Among these PCR-positive samples, 11 (36.7%) were also histologically confirmed as SP. Conversely, nine PCR-negative lesions (29%) were histologically consistent with SP, suggesting either low viral load or the presence of unamplified PV types. The overlap and divergence between molecular and histopathological diagnoses are summarized in Figure 3.

Overall, 39 cases were confirmed via either PCR and/or histopathology as consistent with CPV infection. These lesions predominantly exhibited cauliflower-like macroscopic morphology (61.5%), followed by nodular (20.5%), filiform (10.3%), and pigmented plaques (7.7%). Macroscopically, the lesions identified as papillomatosis or CPV-positive were predominantly cauliflower-like, followed by nodular, filiform, and pigmented plaques (Figure 4).

To explore potential epidemiological and clinical factors associated with viral detection, we performed bivariate analyses comparing PCR positivity with host characteristics (sex, age, and breed), geographic origin, lesion site, lesion morphology, and histopathological diagnosis (Table 2). Regarding host-related variables, no statistically significant differences were observed between sexes (*p* = 0.559), with females comprising 57.1% of PCR-positive cases and males 45.0%. Similarly, age did not significantly influence viral detection (*p* = 0.750); PCR positivity was 50.0% in dogs aged 0–3 years and 55.3% in those older than 3 years. However, the breed group showed a significant association with PCR status (*p* = 0.002). All dogs classified within the small popular breed group (e.g., Poodle, Pinscher, Shih Tzu, and Dachshund) were PCR-positive (10/10), whereas mixed breeds and medium/large popular breeds exhibited lower positivity rates (40.7% and 36.4%, respectively).

A significant geographic association was also identified (*p* = 0.026): dogs from the state of Acre were more likely to be PCR-positive (72.2%) compared to those from Rondônia (39.5%) (*p* = 0.026). In contrast, the anatomical location of lesions (*p* = 0.262), macroscopic morphology (*p* = 0.073), and histopathological classification (*p* = 0.254) did not show statistically significant associations with PCR results. Although not statistically significant, it is noteworthy that all pigmented plaque lesions (3/3) tested PCR-positive, and that cauliflower-like lesions tended to present higher positivity rates (57.6%) compared to nodular (30.8%) and filiform (33.3%) forms.

### 3.3. Histological Analysis of Non-Papillomatous Lesions

Lesions that did not meet the criteria for SP diagnosis (*n* = 22) were histologically classified into various categories. These included sebaceous adenomas, melanocytomas, histiocytomas, chronic otitis, and trichoblastomas, among others. Several samples showed non-specific hyperkeratosis or inflammatory changes, and some were deemed inconclusive or lacking significant histopathological alterations (Table 3).

### 3.4. Phylogenetic Characterization of CPVs

Of the PCR-positive samples, 16 yielded high-quality *L1* sequences suitable for phylogenetic analysis. Most sequences showed >99% nucleotide identity with CPV1, the prototype of the *Lambdapapillomavirus* 2 species (GenBank NC_001619). One notable exception was sample 28AC18BR (PV742302), which exhibited 99% identity with CPV8, a member of the *Chipapillomavirus* 3 species, and 98% identity with a Swiss CPV8 strain (GenBank YP_004857848) associated with pigmented plaques. All other sequences—including 05RO19_BR (PV742297), 07RO19_BR (PV742298), 12RO19_BR (PV742299), 15RO19_BR (PV742301), 17RO19_BR (PV742300), 28RO19_BR (PV742302), 30RO19_BR (PV742303), 30AC18_BR (PQ567121), 32AC19_BR (PQ567122), 41AC19_BR (PQ567123), 42AC19_BR (PQ567124), and 43AC19_BR (PQ567125)—clustered consistently with CPV1.

The phylogenetic tree based on the partial *L1* region of 353 bp (Figure 5) confirmed the taxonomic placement of CPV1 and CPV8 among known *Canis familiaris papillomaviruses*, demonstrating strong bootstrap support for the respective clades.

### 3.5. Whole-Genome Sequencing of Selected Isolates

Three samples (16AC18BR, 20AC18BR, and 34AC19BR) with high DNA quality were selected for whole-genome sequencing via high-throughput sequencing. The complete genomes ranged from 8607 to 8626 bp, and all included the canonical PV gene set: *E1*, *E2*, *E4*, *E5*, *E6*, *E7*, *L1*, *L2*, and a *URR* (upstream regulatory region). Genome coverage depth ranged from 20× to 35×, ensuring high confidence in base calling and genomic structure. All three genomes clustered within *Lambdapapillomavirus* 2 (CPV1), supporting the partial *L1*-based findings. These sequences are deposited in GenBank under the following accession numbers: 16AC18_BR (PQ570013), 20AC18_BR (PQ570014), and 34AC18_BR (PQ570015). The corresponding phylogenetic tree, including these three genomes and reference sequences from the PaVE database, is presented in Figure 6.

## 4. Discussion

Papillomavirus infections in dogs most commonly manifest as oral and cutaneous papillomas, both of which are typically recognizable through clinical examination due to their characteristic exophytic, verrucous appearance [22,23]. In this study, 61 lesions with macroscopic features consistent with papillomatosis were collected from domiciled dogs in the Western Brazilian Amazon, with a predominance of lesions located in the oral cavity and limbs, in line with previous reports on anatomical predilection [23,24].

Although papillomatosis is classically considered a disease of young dogs with immature immune systems [23], our findings challenge this paradigm. Among the 30 dogs confirmed by PCR, only six were ≤3 years of age, while 24 animals were older. The absence of a significant association between age group and PCR suggests that factors beyond age-related immune status, such as individual immunogenetics, co-infections, or environmental pressures, may influence disease expression. This notion aligns with studies demonstrating that many dogs may harbor papillomavirus DNA asymptomatically, with lesion development being contingent on immune dysregulation [22,24]. Similarly, sex was not associated with viral detection, supporting the idea that intrinsic host variables exert limited influence on CPV occurrence in naturally infected dogs.

In contrast, strong associations were identified with breed group and geographic origin. The finding that small popular breeds were consistently PCR-positive may reflect behavioral or immunological characteristics that enhance susceptibility, such as grooming habits or breed-specific immune responses. Interestingly, although more animals were sampled in Rondônia, the state of Acre showed a significantly higher proportion of PCR-positive cases. This may reflect regional differences in viral transmission dynamics, veterinary service access, or even the ability of local veterinarians to clinically recognize papillomatous lesions, potentially leading to more targeted and accurate sample selection in Acre. Together, these findings suggest that certain host and geographic factors—particularly breed groups and state of origin—may influence the likelihood of papillomavirus detection in dogs with papillomatous lesions. Conversely, histological diagnosis and lesion morphology alone were not reliable predictors of PCR positivity, reinforcing the importance of integrating molecular and clinical–pathological approaches for accurate diagnostic assessment.

The overall PCR positivity rate (49.2%) highlights the diagnostic challenges in detecting papillomavirus DNA in clinical samples. Notably, nearly half of the squamous papillomas (SPs) confirmed histologically were PCR-negative. This discrepancy reinforces the importance of combining histopathological evaluation with molecular tools, particularly in lesions with characteristic morphological features. Several factors may account for these results, including low viral load, DNA degradation, uneven viral distribution within the tissue, or the presence of PCR inhibitors. Another key consideration is the use of degenerate FAP59/64 primers, which, despite their broad reactivity, preferentially amplify a limited spectrum of CPV types (1–5, 7), and may fail to detect divergent or low-copy-number variants [3,24].

Beyond technical factors, the “hit-and-run” oncogenic model provides a compelling biological explanation for histologically confirmed squamous papillomas (SPs) lacking detectable viral DNA. In this model, the viral genome is required to initiate—but not maintain—neoplastic transformation [25]. Despite these challenges, FAP59/64 primers remain widely used in epidemiological screening due to their cross-genera utility and reliability in initial virus detection. In fact, the successful amplification of CPV8 in one of the positive cases, despite this genotype being less efficiently targeted by FAP59/64, demonstrates the method’s capacity to occasionally detect uncommon or underrepresented CPV types. Our data also support previous observations that papillomavirus infections in dogs may follow heterogeneous clinical courses, influenced by complex virus–host–environment interactions [3,10], which further underscores the need for a multimodal and context-sensitive diagnostic approach in PV surveillance and research.

To date, 33 distinct CPV types have been described, each demonstrating varying tissue tropism and pathogenic potential [6,9]. In our study, CPV1 was the predominant type detected, aligning with its known association with oral and cutaneous papillomas and its wide global distribution across five continents [26,27,28,29,30,31,32,33,34,35,36,37]. CPV1-positive lesions in this cohort included nodular, filiform, cauliflower-like, and even pigmented plaque morphologies—suggesting the phenotypic versatility of this genotype. This reinforces the concept that CPV1 remains the most ubiquitous and clinically relevant papillomavirus in dogs.

Notably, one sample (28AC18BR) revealed CPV8, a virus traditionally associated with pigmented plaques and cutaneous squamous cell carcinoma, particularly in *Chipapillomavirus* infections [9,28,29]. However, in this case, CPV8 was detected in a filiform SP, which closely resembled classical benign papillomas rather than pigmented plaques. This finding raises important questions regarding the pathogenic spectrum of CPV8 and supports its potential to induce diverse lesion types. Its high sequence identity with a CPV8 strain from Switzerland [YP_004857848] further suggests that this genotype is conserved across distant geographic regions, despite differences in lesion phenotype.

Although no statistically significant associations were detected between PCR positivity and lesion morphology or anatomical location, some trends were noteworthy. The consistent detection of CPVs in pigmented plaques and the relatively high positivity in cauliflower-like lesions suggest that certain morphologies may be more predictive of viral etiology, even if not statistically confirmed in this cohort. These observations highlight the potential diagnostic value of macroscopic evaluation when combined with histopathology and PCR.

The histological features of confirmed SPs in this study—epidermal hyperplasia, marked hyperkeratosis, fibroblast proliferation, and eosinophilic intranuclear inclusions—were consistent with those described in the literature [30,31,32,33]. Lesions were most commonly localized to the lips, oral mucosa, and head, a distribution pattern also reported in studies from southern and northeastern Brazil [30,31].

The sequencing of partial *L1* genes from 16 PCR-positive samples revealed strong identity (>99%) with CPV1, and the three complete genomes obtained confirmed their classification within the *Lambdapapillomavirus* 2 species. The low genetic divergence among these CPV1 isolates—even across geographically distinct Brazilian states—highlights the remarkable genomic stability of this virus. These findings are in agreement with previous molecular surveys conducted in South America [30,31], North America [34,35], Asia [11], Africa [36], and Europe [37,38].

The detection of CPV1 and CPV8 in dogs from the Western Amazon represents the first molecular and phylogenetic characterization of canine papillomaviruses in this biome. Despite the Amazon’s unique ecological complexity, our CPV1 strains were nearly identical to isolates from southern and northeastern Brazil [30,31,39], suggesting either broad viral dissemination or limited mutational drift. This genetic homogeneity raises intriguing questions about the evolutionary constraints and selective pressures acting on CPVs in different environments.

While bovine papillomaviruses (BPVs) have been widely studied and are known for their oncogenic potential and capacity to induce malignant lesions, even in cattle from the amazon region [40], studies on CPVs remain limited—especially in remote, biodiverse areas such as the Amazon. Unlike BPVs, CPVs are typically associated with benign cutaneous and mucosal lesions in dogs; however, the detection of CPV8 in an atypical lesion phenotype in this study challenges existing assumptions and expands the known pathogenic spectrum of these viruses.

In parallel with the viral findings, 22 lesions did not meet the histological criteria for squamous papillomas and were classified as non-viral pathologies. The most common diagnosis was melanocytoma, followed by other benign neoplasms such as sebaceous adenomas, trichoblastoma, and histiocytoma, all exhibiting features distinct from PV-induced lesions [41,42,43]. Inflammatory conditions like chronic otitis and ulcerative stomatitis were also identified, reflecting the complexity of differential diagnoses in exophytic lesions [44,45]. Additionally, some samples showed no significant alterations or were inconclusive, possibly due to regressed lesions, non-viral causes, or technical limitations during tissue processing. These findings highlight the relevance of histopathological analysis in differentiating viral papillomatosis from other dermatological conditions.

Taken together, our findings underscore the multifactorial nature of CPV infection in dogs and suggest that viral detection and lesion development result from a complex interplay between host, pathogen, and environment. The identification of epidemiological predictors such as breed and location, alongside the variable performance of PCR across lesion types, calls for an integrated diagnostic framework that combines clinical, histopathological, and molecular tools.

In the context of the Western Amazon—an ecologically rich but understudied region—this study provides novel insights into CPV diversity, distribution, and diagnostic challenges. Future research should prioritize the detection of novel CPV types, the development of broader-range primers, and the investigation of viral dynamics across breeds and ecological niches. Such efforts are crucial to improving diagnosis, surveillance, and our understanding of papillomavirus evolution in domestic animals.

## 5. Conclusions

This study reinforces the value of integrating molecular and histopathological approaches to improve the accuracy of CPV diagnosis. Our findings expand the epidemiological landscape of CPVs by documenting the presence of CPV1 and CPV8 in dogs from the Brazilian Amazon and underscore the need for continued viral surveillance in underrepresented regions. By illuminating the genetic stability of CPV1 and detecting CPV8 in an unusual lesion phenotype, this study provides new insights into papillomavirus diversity and evolution in companion animals. Understanding these dynamics is crucial not only for veterinary virology, but also for comparative oncology and One Health strategies.

## Figures and Tables

**Figure 1 microorganisms-13-01811-f001:**
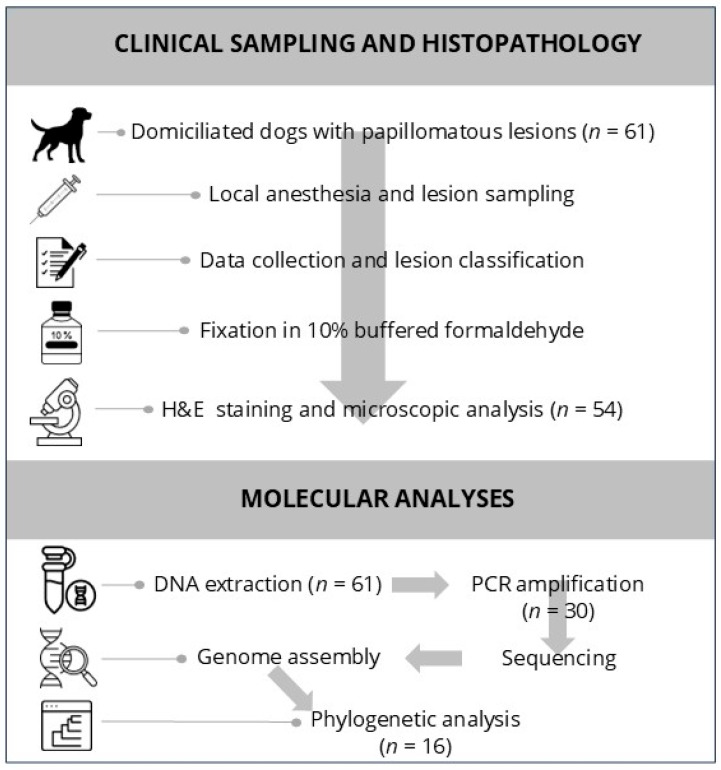
Workflow diagram summarizing the methodology used in the study. (**Top panel**): Clinical sampling and histopathological evaluation of 61 dogs with papillomatous lesions, including lesion excision, classification, and histological analysis. (**Bottom panel**): Molecular workflow, from DNA extraction and PCR amplification to genome assembly and phylogenetic analysis.

**Figure 2 microorganisms-13-01811-f002:**
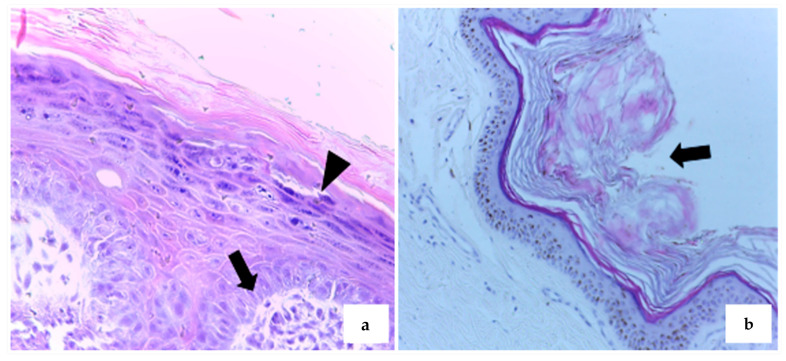
Exophytic papillomatous proliferation of the epithelium in dogs sampled in the study. (**a**) Fibroblast proliferation in the superficial dermis (arrow), associated with hyperplasia, hyperkeratosis, and hypergranulosis (arrowhead) in the underlying epidermis. HE, 20×. (**b**) Hyperkeratosis (arrow) covering hyperplastic epidermis. HE, 10×.

**Figure 3 microorganisms-13-01811-f003:**
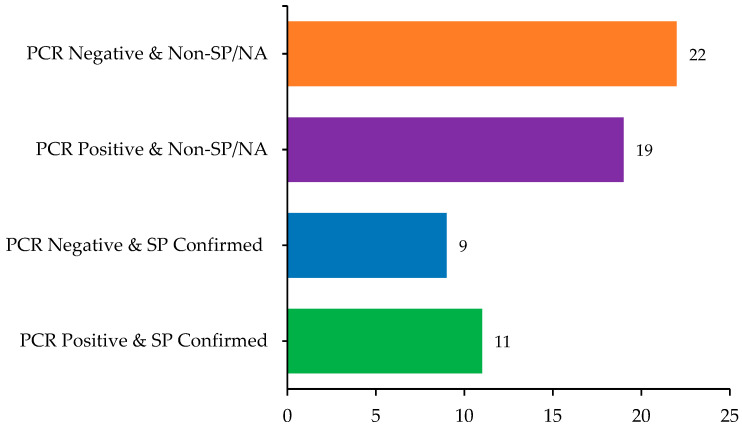
Comparison of histopathological and molecular findings in sampled dogs (*n* = 61), highlighting overlaps and discrepancies between PCR results and squamous papilloma (SP) diagnosis.

**Figure 4 microorganisms-13-01811-f004:**
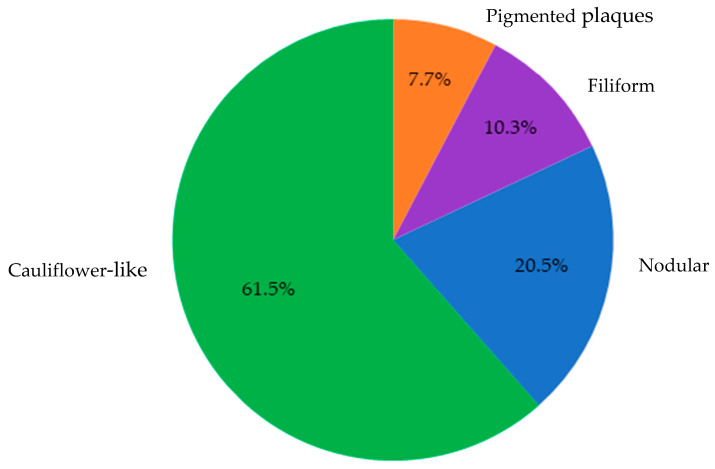
Distribution of lesion morphologies among samples diagnosed as papillomatosis via PCR and/or histopathology (*n* = 39). Cauliflower-like lesions predominated (61.5%), followed by nodular (20.5%), filiform (10.3%), and pigmented plaques (7.7%).

**Figure 5 microorganisms-13-01811-f005:**
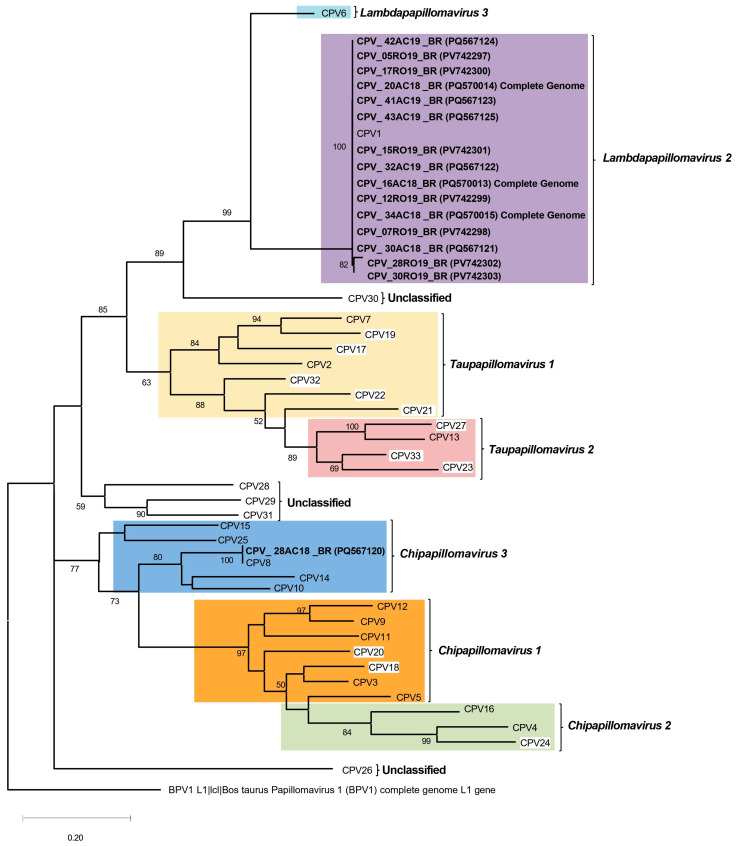
Phylogenetic tree of partial *L1* region, which reveals major clades representing different species of *Canis familiaris papillomavirus* types, with an outgroup of *Colobus guereza papillomavirus* generated using MEGA (version 5.0) software. The sequences generated herein are bold. The three CPV1 isolates sequenced as complete genomes (16AC18_BR, 20AC18_BR, and 34AC18_BR) are also included in the tree and are indicated in bold font. CPVs with white framing are those not classified within species at PaVE.

**Figure 6 microorganisms-13-01811-f006:**
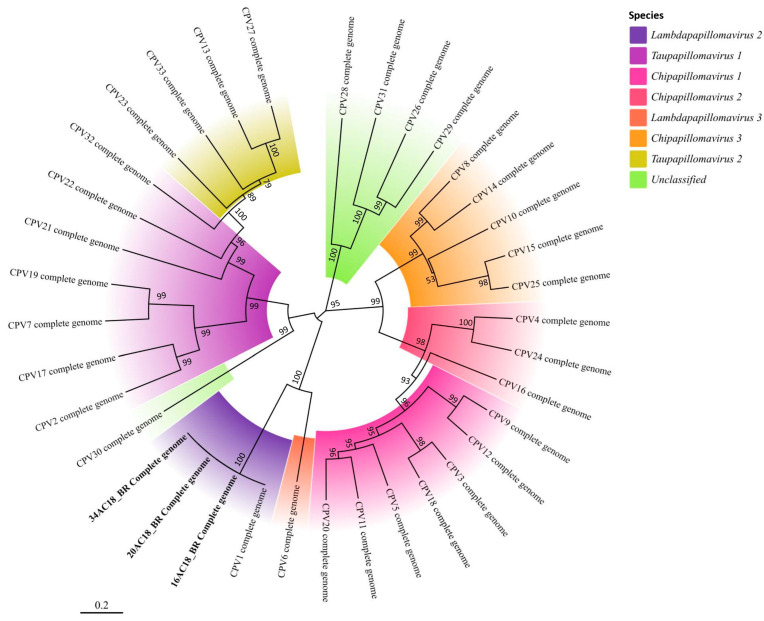
Phylogenetic tree based on the complete genome sequences of *Canis familiaris papillomaviruses* (CPVs), constructed using the maximum likelihood method with 1000 bootstrap replicates. The tree includes reference genomes from GenBank and the three CPV1 genomes sequenced in this study (16AC18_BR, 20AC18_BR, and 34AC18_BR) highlighted in bold.

**Table 1 microorganisms-13-01811-t001:** Complete samples data, including ID, PCR result, age (Y: year/s, M: month/s), breed, municipality/state, anatomic location, and macroscopy/microscopy features.

ID	PCR	Age	Sex	Breed	Municipality/State	Anatomic Location	Macroscopy	Microscopy
16AC18BR	CPV1	5Y	F	Poodle	Rio Branco/Acre	Ear	Nodular	NA ^2^
20AC18BR	CPV1	8Y	F	American Bully	Rio Branco/Acre	Eye	Nodular	SP ^3^
27AC18BR	(+) ^1^	4Y	F	Pinscher	Rio Branco/Acre	Limb	Nodular	NA
28AC18BR	CPV8	4Y	F	Mixed breed	Rio Branco/Acre	Limb	Filiform	SP
30AC18BR	CPV1	2Y	F	Pinscher	Rio Branco/Acre	Head	Cauliflower	SP
32AC19BR	CPV1	7Y	F	Mixed breed	Rio Branco/Acre	Oral	Cauliflower	SP
34AC19BR	CPV1	1Y	F	Mixed breed	Rio Branco/Acre	Oral	Cauliflower	SP
35AC19BR	(+)	12Y	M	Pitbull	Rio Branco/Acre	Limb	Filiform	NA
41AC19BR	CPV1	2Y	M	Poodle	Rio Branco/Acre	Oral	Cauliflower	NA
42AC19BR	CPV1	2Y	F	Shih Tzu	Rio Branco/Acre	Oral	Cauliflower	NA
43AC19BR	CPV1	4Y	M	Mixed breed	Rio Branco/Acre	Eye	Cauliflower	NA
19AC18BR	(-)	8Y	F	Mixed breed	Rio Branco/Acre	Teat	Nodular	SP
21RO18BR	(-)	4 M	M	Labrador	Porto Velho/Rondônia	Oral	Nodular	SP
22RO18BR	(-)	8Y	M	Labrador	Porto Velho/Rondônia	Oral	Nodular	ND ^4^
24AC18BR	(-)	8Y	M	Mixed breed	Rio Branco/Acre	Ear	Cauliflower	ND
25AC18BR	(-)	3Y	M	Mixed breed	Rio Branco/Acre	Abdomen	Nodular	SP
26AC18BR	(-)	3Y	F	Mixed breed	Rio Branco/Acre	Teat	Nodular	SP
31AC18BR	(-)	10Y	F	Mixed breed	Rio Branco/Acre	Teat	Filiform	ND
01RO19BR	(+)	12Y	M	Mixed breed	Jarú/Rondônia	Oral	Cauliflower	SP
02RO19BR	(-)	8Y	F	Pitbull	Ji-Paraná/Rondônia	Neck	Filiform	ND
03RO19BR	(+)	13Y	F	Poodle	Ji-Paraná/Rondônia	Back	Pigmented plaques	ND
04RO19BR	(+)	12Y	F	Boxer	Ji-Paraná/Rondônia	Limb	Pigmented plaques	ND
05RO19BR	CPV1	8M	M	Mixed breed	Ouro-Preto/Rondônia	Oral	Cauliflower	ND
06RO19BR	(-)	8M	M	Pitbull	Ji-Paraná/Rondônia	Oral	Cauliflower	SP
07RO19BR	CPV1	3Y	M	Mixed breed	Ji-Paraná/Rondônia	Oral	Cauliflower	SP
08RO19BR	(-)	10Y	M	Basset Hound	Ji-Paraná/Rondônia	Head	Cauliflower	SP
09RO19BR	(-)	10Y	M	Pitbull	Ji-Paraná/Rondônia	Limb	Filiform	ND
10RO19BR	(-)	13Y	M	Mixed breed	Ji-Paraná/Rondônia	Limb	Cauliflower	SP
11RO19BR	(-)	10Y	M	Mixed breed	Ji-Paraná/Rondônia	Ear	Cauliflower	ND
12RO19BR	CPV1	8Y	F	Mixed breed	Nova-Londrina/Rondônia	Oral	Cauliflower	SP
13RO19BR	(-)	2Y	M	Pitbull	Nova-Londrina/Rondônia	Oral	Cauliflower	SP
14RO19BR	(+)	13Y	F	Mixed breed	Ji-Paraná/Rondônia	Oral	Cauliflower	ND
15RO19BR	CPV1	13Y	M	Pitbull	Ji-Paraná/Rondônia	Chest	Cauliflower	ND
16RO19BR	(-)	12Y	M	Mixed breed	Ji-Paraná/Rondônia	Oral	Cauliflower	SP
17RO19BR	CPV1	7Y	F	Dachshund	Ji-Paraná/Rondônia	Limb	Cauliflower	ND
18RO19BR	(-)	13Y	F	Mixed breed	Ji-Paraná/Rondônia	Oral	Cauliflower	ND
19RO19BR	(-)	12Y	F	Mixed breed	Ji-Paraná/Rondônia	Eye	Filiform	ND
20RO19BR	(-)	12Y	F	Mixed breed	Ji-Paraná/Rondônia	Teat	Filiform	ND
21RO19BR	(-)	12Y	F	Mixed breed	Ji-Paraná/Rondônia	Limb	Cauliflower	ND
22RO19BR	(-)	MI ^5^	MI	MI	Ji-Paraná/Rondônia	Back	Cauliflower	ND
23RO19BR	(-)	MI	MI	MI	Ji-Paraná/Rondônia	Oral	Nodular	ND
24RO19BR	(-)	MI	MI	MI	Ji-Paraná/Rondônia	Eye	Nodular	ND
25RO19BR	(-)	MI	MI	MI	Ji-Paraná/Rondônia	Limb	Nodular	ND
26RO19BR	(-)	MI	MI	MI	Ji-Paraná/Rondônia	Abdomen	Nodular	ND
27RO19BR	(+)	MI	MI	MI	Ji-Paraná/Rondônia	Oral	Cauliflower	SP
28RO19BR	CPV1	MI	MI	MI	Ji-Paraná/Rondônia	Neck	Cauliflower	ND
30RO19BR	CPV1	MI	MI	MI	Ji-Paraná/Rondônia	Oral	Cauliflower	SP
31RO19BR	(-)	MI	MI	MI	Ji-Paraná/Rondônia	Oral	Cauliflower	ND
32RO19BR	(-)	MI	MI	MI	Ji-Paraná/Rondônia	Chest	Filiform	ND
33RO19BR	(+)	7Y	F	Shih Tzu	Ji-Paraná/Rondônia	Eye	Nodular	ND
34RO19BR	(-)	MI	MI	MI	Ji-Paraná/Rondônia	Teat	Filiform	ND
35RO19BR	(-)	8Y	F	Mixed breed	Ji-Paraná/Rondônia	Oral	Cauliflower	ND
36RO19BR	(-)	10Y	F	Mixed breed	Ji-Paraná/Rondônia	Neck	Cauliflower	ND
39RO19BR	(-)	5M	F	Mixed breed	Ji-Paraná/Rondônia	Oral	Cauliflower	ND
40RO19BR	(+)	11Y	F	Poodle	Ji-Paraná/Rondônia	Chest	Filiform	ND
41RO19BR	(+)	8Y	M	Mixed breed	Ji-Paraná/Rondônia	Eye	Filiform	ND
42RO19BR	(-)	9Y	F	Mixed breed	Ji-Paraná/Rondônia	Teat	Filiform	ND
43RO19BR	(+)	11Y	M	Mixed breed	Ji-Paraná/Rondônia	Ear	Cauliflower	ND
44RO19BR	(+)	8Y	MI	MI	Ji-Paraná/Rondônia	Oral	Cauliflower	SP
45AC19BR	(+)	6Y	MI	MI	Rio Branco/Acre	Oral	Cauliflower	ND
02AC23BR	(+)	7Y	F	Pinscher	Rio Branco/Acre	Vulva	Pigmented plaques	NA

^1^ (+) = sample tested positive via PCR but was not sequenced; ^2^ NA = sample not analyzed due to insufficient tissue; ^3^ SP = squamous papilloma; ^4^ ND = non-diagnostic to SP; and ^5^ MI = missing information; (-) =. sample tested negative via PCR.

**Table 2 microorganisms-13-01811-t002:** Association between clinical–epidemiological variables and PCR detection of canine papillomavirus DNA in dogs with papillomatous lesions.

Variables		Viral Status (PCR)		
		Positive (%)	Negative (%)	*p*-Value
**Sex**	Female	16 (57.1)	12 (42.9)	0.559
	Male	9 (45.0)	11 (55.0)	
**Age**	0–3 years	6 (50.0)	6 (50.0)	0.750
	>3 years	21 (55.3)	17 (44.7)	
**Breed ***	Mixed breed	11 (40.7)	16 (59.3)	0.002
	Small popular	10 (100.0)	0 (0.0)	
	Medium/large popular	4 (36.4)	7 (63.6)	
**State**	Acre	13 (72.2)	5 (27.8)	0.026
	Rondônia	17 (39.5)	26 (60.5)	
**Anatomic localization**	Head	21 (55.3)	17 (44.7)	0.262
	Torso	4 (28.6)	10 (71.4)	
	Limb	5 (55.6)	4 (44.4)	
**Morphology of the lesions ***	Cauliflower	19 (57.6)	14 (42.4)	0.073
	Nodular	4 (30.8)	9 (69.2)	
	Filiform	7 (33.3)	8 (66.7)	
	Pigmented plaques	3 (100.0)	0 (0.0)	
**Microscopy**	Squamous papilloma (SP)	11 (55.0)	9 (45.0)	0.254
	Non-diagnostic to SP	12 (35.3)	22 (64.7)	

*p*-values were obtained using Fisher’s exact test for 2 × 2 contingency tables. * Monte Carlo simulation with 10,000 replications was employed for larger tables. Statistical significance was set at *p* < 0.05.

**Table 3 microorganisms-13-01811-t003:** Diagnostic summary and microscopic characterization of canine skin and mucosal lesions not diagnosed as squamous papilloma.

Sample	Diagnosis	Histopathological Description
22RO18BR	Sebaceous hyperplasia	Sample composed of skin with sebaceous gland hyperplasia and epidermal acanthosis.
24AC18BR	Histiocytoma	Non-encapsulated dermal neoplastic proliferation of round mesenchymal cells, with eosinophilic cytoplasm, prominent nucleoli, moderate pleomorphism, and high mitotic activity (≈30 mitoses/10 HPF).
31AC18BR	Dermatofibrosis	Collagen proliferation and infiltrate of mast cells and lymphocytes.
02RO19BR	Collagenous hamartoma	Papillary exophytic nodule composed of marked disorganized collagen proliferation, covered by thin stratified keratinized squamous epithelium.
09RO19BR	Sebaceous adenoma	Well-differentiated, non-encapsulated sebocytic proliferation in small lobules with scant collagenous stroma. Polygonal cells with vacuolated cytoplasm, round nuclei, stippled chromatin, and prominent nucleoli. Mild anisocytosis and anisokaryosis, no mitoses.
11RO19BR	Chronic otitis	Epidermis with marked acanthosis and dermal inflammatory infiltrate composed of lymphocytes, plasma cells, macrophages, and neutrophils. Infiltrate also involves dilated ceruminous glands with eosinophilic material and neutrophils. Moderate proliferation of fibrous connective tissue in the dermis.
18RO19BR	Melanocytoma	Poorly defined, non-encapsulated melanocytic neoplastic proliferation with individual polygonal cells, cytoplasm containing melanin granules, and mild anisocytosis. No mitoses. Numerous freezing artifacts.
19RO19BR	Melanocytoma	Poorly defined, non-encapsulated melanocytic neoplastic proliferation of polygonal cells with melanin granules often obscuring the nuclei. Nuclei are round to oval with finely stippled chromatin and prominent nucleoli. Mild anisocytosis and anisokaryosis, no mitotic figures.
20RO19BR	Melanocytoma	Poorly defined, non-encapsulated melanocytic neoplastic proliferation of polygonal cells with melanin granules often obscuring the nuclei. Nuclei are round to oval, finely stippled chromatin, prominent nucleoli. Mild anisocytosis and anisokaryosis, no mitoses observed.
21RO19BR	Inconclusive	Moderate multifocal epidermal acanthosis associated with moderate orthokeratotic hyperkeratosis.
22RO19BR	No significant histopathological lesions	Mild multifocal orthokeratotic hyperkeratosis.
23RO19BR	No significant histopathological lesions	Mild multifocal orthokeratotic hyperkeratosis.
24RO19BR	Melanocytoma	Poorly defined, non-encapsulated melanocytic proliferation of individual polygonal cells with cytoplasm containing melanin. Round to oval nuclei with stippled chromatin and prominent nucleoli. Mild anisocytosis and anisokaryosis, no mitoses. Moderate inflammatory infiltrates of lymphocytes, plasma cells, and macrophages.
25RO19BR	Sebaceous adenoma	Well-differentiated, non-encapsulated sebocytic proliferation in small lobules with scant stroma. Polygonal cells with vacuolated cytoplasm, round nuclei, and prominent nucleoli. Mild anisocytosis and anisokaryosis, no mitoses. Epidermis with moderate acanthosis.
26RO19BR	Inconclusive	Deep dermis with focal extensive area of dilated blood vessels filled with erythrocytes, surrounded by moderate inflammatory infiltrates of lymphocytes, plasma cells, and macrophages.
31RO19BR	Chronic active ulcerative stomatitis	Submucosa with marked inflammatory infiltrates (neutrophils, lymphocytes, plasma cells, and macrophages), associated with fibrovascular proliferation and fibrin deposition. Mucosa shows multifocal ulceration with neutrophilic infiltrate, bacterial aggregates, and fibrin. Remaining epithelium is moderately hyperplastic with digitiform projections into the submucosa.
32RO19BR	Trichoblastoma	Moderately defined epithelial neoplastic proliferation in superficial dermis, with polygonal cells in trabeculae over abundant fibrocollagenous stroma. Oval to elongated nuclei, stippled chromatin, prominent nucleoli, mild anisocytosis and anisokaryosis, and 25 mitoses in 2.37 mm^2^. Mild lymphoplasmacytic infiltrate in underlying dermis.
34RO19BR	No significant histopathological lesions	Mild multifocal orthokeratotic hyperkeratosis.
35RO19BR	No significant histopathological lesions	Mild multifocal orthokeratotic hyperkeratosis. In the dermis around skin adnexa, mild infiltrates of lymphocytes, plasma cells, and macrophages.
36RO19BR	No significant histopathological lesions	Marked diffuse orthokeratotic hyperkeratosis associated with mild epithelial acanthosis.
39RO19BR	No significant histopathological lesions	Mild infiltrate of lymphocytes and macrophages, along with mild fibrosis.
42RO19BR	Melanoma	Poorly defined melanocytic neoplastic proliferation in dermis, with polygonal cells containing melanin that obscures nuclei. Nuclei are round to oval, with moderate anisocytosis and anisokaryosis, and 20 mitoses. Moderate inflammatory infiltrate present.

## Data Availability

The original contributions presented in this study are included in the article. Further inquiries can be directed to the corresponding authors.

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
