# Peer review of "Unveiling the Genetic Landscape of Canine Papillomavirus in the Brazilian Amazon"

_microorganisms, 2025, doi:10.3390/microorganisms13081811_

Round 1
Reviewer 1 Report
Comments and Suggestions for Authors
This manuscript reports the existence and types of canine papillomaviruses (CPVs) in domiciled dogs from two states of Brazil. Among 61 samples collected, the authors detected 30 CPV positives using PCR, yielded 16 high-quality L1 sequences for phylogenetic analysis, and got three complete CPV genomic sequences. Although the manuscript is well written, the analyses were quite limited that could hardly provide reliable information about the prevalence of CPVs in the samples.
- The detection of CPV in this study mainly depended on general PCR using primers FAP59 and FAP64. However, this pair of primers were designed to detect human papillomaviral DNA more than 20 years ago (in 1999). Thus, the effectiveness and accuracy of using these primers to detect CPV DNA should be verified. The alignment of the primers and L1 ORF sequences of different PVs is a typical verification suggested to be conducted. According to the current results, the primers might not be suitable for the detection, since less than half L1 sequences were yielded for phylogenetic analysis, which may lead to bias in revealing the prevalent types of CPVs. Therefore, new primers better for CPV detection may be designed.
- Since the histopathological features of the samples are quite diverse, the correlation analysis of PV types and histopathological features is suggested, which will provide more critical information about the pathogenicity of CPVs.
- More epidemiology analysis is suggested to reveal the transmission of CPVs in Brazil.
Author Response
Dear Reviewer 1,
Thank you for your thoughtful review and for dedicating time to evaluate our manuscript. We sincerely appreciate your constructive feedback, which has significantly strengthened the scientific rigor of our work. Below, we address your central concerns regarding the scope of our article:
Comment 1:
The detection of CPV in this study mainly depended on general PCR using primers FAP59 and FAP64. However, this pair of primers were designed to detect human papillomaviral DNA more than 20 years ago. The effectiveness and accuracy of using these primers to detect CPV DNA should be verified. The alignment of the primers and L1 ORF sequences of different PVs is a typical verification suggested to be conducted. According to the current results, the primers might not be suitable for the detection, since less than half L1 sequences were yielded for phylogenetic analysis, which may lead to bias in revealing the prevalent types of CPVs. Therefore, new primers better for CPV detection may be designed.
Response:
We appreciate the reviewer’s insightful comment. We fully recognize the historical context and limitations of the FAP59/64 primer set, which was originally designed for detecting mucosal HPV DNA. However, it is important to emphasize that this primer pair remains one of the most commonly used tools for screening papillomavirus infections across a broad range of host species, including canines, and has been validated in numerous CPV molecular surveys (e.g., de Alcântara et al., 2014[1]). Despite its preferential amplification of CPV types 1–5 and 7, FAP59/64 has demonstrated satisfactory performance in epidemiological studies where initial broad detection is the primary aim.
In the context of our study focused on the initial molecular mapping of CPVs in the Western Amazon FAP59/64 was employed as a screening tool to detect and genotype the most prevalent circulating viruses. Indeed, the successful amplification of CPV8 in one of the positive cases, despite this genotype being suboptimally targeted by the primer set, further demonstrates its utility in occasionally capturing divergent or underrepresented CPV types.
To address the reviewer’s suggestion, we have now included a brief discussion on this topic (Discussion, lines 317–325), emphasizing both the limitations and justifications for the use of FAP59/64 in this screening study.
We agree that future studies should explore the design and validation of new primer sets tailored to the detection of divergent CPV genotypes, and we highlight this recommendation in the final paragraph of the Discussion.
- de Alcântara, B.K.; Alfieri, A.A.; Rodrigues, W.B.; Otonel, R.A.A.; Lunardi, M.; Headley, S.A.; Alfieri, A.F. Identification of Canine Papillomavirus Type 1 (CPV1) DNA in Dogs with Cutaneous Papillomatosis. Vet. Bras. 2014, 34, 1223–1226. https://doi.org/10.1590/S0100-736X2014001200005.
Comment 2 and 3:
Since the histopathological features of the samples are quite diverse, the correlation analysis of PV types and histopathological features is suggested, which will provide more critical information about the pathogenicity of CPVs. More epidemiology analysis is suggested to reveal the transmission of CPVs in Brazil.
Response:
We performed additional statistical analyses to explore potential associations between the presence of papillomavirus DNA (PCR status) and histopathological classification, as well as other clinical and anatomical variables. Specifically, we assessed the correlation between PCR results and histological diagnosis (squamous papilloma vs. non-diagnostic) using Fisher’s Exact Test, which is well-suited for 2×2 contingency tables with small sample sizes and sparse data.
To broaden our understanding of the pathogenic and epidemiological profile of CPVs, we also examined other variables such as macroscopic lesion morphology, anatomical localization, breed group, age, sex, and geographic origin. For multi-category variables, exact p-values were estimated using Monte Carlo simulations with 10,000 replications. This method allows for accurate estimation in higher-dimensional tables where standard Fisher’s test is not feasible.
These analyses revealed significant associations with breed group and geographic origin, providing insights into potential host and environmental factors that may influence CPV susceptibility and expression. The results have been incorporated into Section 3.2 of the revised manuscript, and their implications are discussed in the revised Discussion section.
We sincerely thank the reviewer for their valuable input. Their comments significantly improved the manuscript. We hope the revised version meets the standards of Microorganisms and we look forward to your final decision.
Best regrads,
Prof. Dr Felipe Masiero Salvarani and Profa. Dra Cíntia Daudt

Reviewer 2 Report
Comments and Suggestions for Authors
In this paper, de Macêdo Sousa and colleagues analysed CPV types present in oral and cutaneous papillomatous lesions in domiciled dogs from two regions in Brazil. For each lesion, authors confirmed by either PCR and/or histopathology and classified the lesion. In addition, they performed a phylogenetic study by sequencing all the isolates.
The paper is well organized and well describes the materials and methods section. Results are well represented, with the right number of figures and tables. Discussions are consistent with results.
Author Response
Dear Reviewer 2,
Thank you for your generous assessment of our manuscript and for recognizing the academic and scientific value of our work on canine papillomaviruses in Brazilian dogs. We particularly appreciate your acknowledgment of the manuscript's organization, methodological clarity, and consistent discussion.
Comment:
In this paper, de Macêdo Sousa and colleagues analysed CPV types present in oral and cutaneous papillomatous lesions in domiciled dogs from two regions in Brazil. For each lesion, authors confirmed by either PCR and/or histopathology and classified the lesion. In addition, they performed a phylogenetic study by sequencing all the isolates.
The paper is well organized and well describes the materials and methods section. Results are well represented, with the right number of figures and tables. Discussions are consistent with results.
Response:
We sincerely thank the reviewer for their positive and encouraging evaluation of our work. We greatly appreciate the recognition of the manuscript’s organization, clarity of the methods section, and consistency of the discussion with the presented results. Constructive feedback such as yours reinforces the value of our findings and motivates us to continue improving and expanding this line of research. We are glad that the study was considered well-structured, and we have carefully reviewed the manuscript again to ensure that all sections remain coherent, informative, and scientifically sound. Thank you for your thoughtful comments and support.
Best regards,
Prof. Dr Felipe Masiero Salvarani and Profa. Dra. Cíntia Daudt

Reviewer 3 Report
Comments and Suggestions for Authors
This study presents interesting reports concerning a canine virus, which is firstly identified and described in the region of the Amazon. However, despite the important data produced, they are underestimated based on the absence of sufficient statistical analyses. My criticism concerns two major issues, which, nevertheless, can be corrected and revised:
1. Since so many clinical data were obtained, why did no the authors try to correlate them statistically with the prevalence of the virus and lesions? This could result to potential risk factors such as age or breed and give the study a better perspective
2. Since the whole genome was sequenced in three samples, why not use these sequences for phylogenetic analyses. So much info is lost this way
There really also some other questions and comments raised that should be enlightened, as follows
What about the remaining 31 that were not tested positive? Any assumption for the etiological agent?
It is not relevant to refer to biodiversity since only dogs were examined. I recommend to erase these parts from the Introduction
In sampling description, in section 2.1, more details are needed. The dog owners brought them to the vet for diagnosis and is this how the lesions were observed?
In section 3.4 please mention how many base pairs were used for phylogenetic analysis
I cannot understand, why not using the three whole genomes for phylogeny reconstruction? Were they submitted in the GenBank? If not they have to be submitted. The entire 3.5 section is meaningless as it is
Author Response
Dear Reviewer 3,
Thank you for your thoughtful review and for dedicating time to evaluate our manuscript. We sincerely appreciate your constructive feedback, which has significantly strengthened the scientific rigor of our work. Below, we address your central concerns regarding the scope of our article:
Comment 1:
Since so many clinical data were obtained, why did no the authors try to correlate them statistically with the prevalence of the virus and lesions? This could result to potential risk factors such as age or breed and give the study a better perspective.
Response:
We appreciate the reviewer’s suggestion and fully agree with its relevance. In the revised version of the manuscript, we have now included a comprehensive statistical analysis to explore possible associations between canine papillomavirus (CPV) detection (PCR positivity) and the clinical-epidemiological variables collected, including sex, age group, breed category, anatomical site, lesion morphology, histopathological classification, and geographic origin.
To allow for robust analysis, some variables were grouped (e.g., age into two categories, breeds into three groups, anatomical sites) to ensure meaningful comparisons based on observed distributions. We applied Fisher’s Exact Test for 2×2 contingency tables and Monte Carlo simulation (10,000 replications) for larger tables, following established statistical practices. The analyses were performed using R software (version cited in the manuscript).
As now presented in Section 3.2 and Table 2, significant associations were observed for breed group (p = 0.002) and state of origin (p = 0.026), suggesting that both host-related and environmental factors may influence CPV prevalence. These findings are further interpreted and contextualized in the Discussion section, providing new insights into potential risk factors and enhancing the epidemiological relevance of the study.
Comment 2:
Since the whole genome was sequenced in three samples, why not use these sequences for phylogenetic analyses. So much info is lost this way.
Response:
We clarify that the three complete genomes obtained in this study were already represented in the original phylogenetic tree, which was constructed using the partial L1 gene — the gold-standard region for papillomavirus classification, as defined by the Papillomavirus Episteme (PaVE) and ICTV. The L1 gene encodes the major capsid protein and is the most conserved and taxonomically informative region, serving as the primary marker for species demarcation among papillomaviruses.
In response to the reviewer’s suggestion, we additionally constructed a separate phylogenetic tree using only the complete genome sequences (accession numbers PQ570013, PQ570014, and PQ570015). This tree yielded a topology that was highly similar to that of the L1-based tree and confirmed the classification of all three isolates within the Lambdapapillomavirus 2 species (CPV1). Given the redundancy in structure and interpretation, we opted not to include this second tree in the main manuscript. However, to ensure transparency, we have attached the full-genome phylogenetic tree as a supplementary figure to this response for the reviewer’s reference.
We appreciate the reviewer’s recommendation, which allowed us to further validate and reinforce the phylogenetic consistency of our findings using both targeted and whole-genome approaches.
Figure 1. Phylogenetic tree based on the complete genome sequences of Canis familiaris papillomaviruses (CPVs), constructed using the Maximum Likelihood method with 1,000 bootstrap replicates. The tree includes reference genomes from GenBank and the three CPV1 genomes sequenced in this study (16AC18_BR, 20AC18_BR, and 34AC18_BR), highlighted in bold and annotated with their GenBank accession numbers. (attached).
Comment 3:
What about the remaining 31 that were not tested positive? Any assumption for the etiological agent?
Response:
We thank the reviewer for this important point. As detailed in Table 2 (page 9) of the revised manuscript, we included the histopathological findings for samples that tested negative by both PCR and histopathology for squamous papillomas. These samples were carefully analyzed, and a variety of alternative diagnoses were established, including benign neoplasms (e.g., melanocytomas, trichoblastomas, histiocytomas), inflammatory conditions (e.g., chronic otitis, ulcerative stomatitis), and some inconclusive cases due to technical limitations.
Additionally, we expanded the Discussion section (lines 374–383) to explore these alternative etiologies and their implications for differential diagnosis. We emphasized the importance of thorough histopathological assessment and the challenges of relying solely on macroscopic features or PCR for diagnosing papillomavirus infections. These revisions help clarify the diverse nature of the lesions and support the need for integrative diagnostic approaches in regions with high environmental and pathological complexity.
Comment 4:
It is not relevant to refer to biodiversity since only dogs were examined. I recommend to erase these parts from the Introduction.
Response:
Thank you for this observation. We understand the concern regarding the relevance of referencing biodiversity when the study population comprises only dogs. However, we respectfully believe that the ecological context of the Amazon biome remains pertinent to the rationale of our research. The Amazon’s exceptional biodiversity and environmental complexity may influence virus–host dynamics in ways not observed in more urbanized or temperate areas. Factors such as host genetic variability, co-infections with endemic pathogens, and environmental pressures (e.g., UV exposure, humidity, or microbiota diversity) could modulate lesion development, viral persistence, or tropism.
Nonetheless, to ensure clarity and focus, we have revised the Introduction to minimize overgeneralization and to better contextualize the mention of biodiversity specifically in relation to potential environmental influences on CPV diversity, evolution, or disease expression. The emphasis is now placed on the uniqueness of the Amazon as a potentially underexplored region for viral surveillance, rather than on its overall biodiversity.
We hope this targeted revision addresses the concern while preserving the scientific rationale for situating our research within this biome.
Comment 5:
Section 2.1 needs more detail on how animals were selected.
Response:
We have revised Section 2.1 to provide a more detailed explanation of the sampling process (lines 84–89). Specifically, we clarified that all dogs included in the study were domiciled animals voluntarily brought by their owners to private and university-affiliated veterinary clinics in the states of Acre and Rondônia between 2018 and 2019. The animals presented with lesions of exophytic or verrucous appearance, and upon clinical evaluation by licensed veterinarians, papillomatous lesions were suspected.
Only dogs with owner consent and lesions consistent with papillomatosis were included. The veterinary professionals performed a clinical assessment and collected data on signalment and lesion characteristics before proceeding to sample collection. These clarifications were added to ensure the reproducibility and transparency of the sampling methodology.
.Comment 6:
In section 3.4 please mention how many base pairs were used for phylogenetic analysis.
Response:
Thank you for your observation. We have now clarified in Section 3.4 (line 252) that the phylogenetic analysis was conducted using the entire 353 base pair (bp) fragment of the L1 gene. This region is widely used for papillomavirus genotyping and comparative analysis in molecular epidemiological studies. The corresponding revision has been made in the manuscript to explicitly state the length of the sequence used.
Comment 7:
I cannot understand, why not using the three whole genomes for phylogeny reconstruction? Were they submitted in the GenBank? If not they have to be submitted. The entire 3.5 section is meaningless as it is.
Response:
The three complete genomes generated in this study (16AC18BR, 20AC18BR, and 34AC19BR) were indeed submitted to GenBank, and their accession numbers (PQ570013, PQ570014, and PQ570015) are now explicitly provided in Section 3.5 of the revised manuscript.
As noted in our response to Comment 2, these genomes were already represented in the original phylogenetic tree through their corresponding partial L1 gene sequences, amplified using the FAP59/64 primers (353 bp).
We thank the reviewer for highlighting this opportunity to strengthen the manuscript and believe the revised section now provides a more robust contribution to CPV molecular epidemiology.
We sincerely thank the reviewer for their valuable input. Their comments significantly improved the manuscript. We hope the revised version meets the standards of Microorganisms and we look forward to your final decision.
Best regards
Prof. Dr. Felipe Masiero Salvarani and Profa. Dra. Cíntia Daudt

Round 2
Reviewer 1 Report
Comments and Suggestions for Authors
All of my questions have been addressed. Thank you for the revision!
Author Response
Dear Reviewer 1,
Thank you very much for your thorough review of our manuscript during the first round of revisions. We sincerely appreciate the time, expertise, and thoughtful insights you dedicated to evaluating our work.
We are delighted to learn that "all of your questions have been addressed" in the revised version. This outcome was made possible entirely due to your constructive feedback, detailed comments, and critical suggestions, which guided our revisions and significantly strengthened the manuscript. Your expertise not only enhanced the clarity and rigor of our work but also deepened its scholarly contribution.
Once again, we extend our deepest gratitude for your invaluable input. We are honored to have benefited from your guidance.
Sincerely,
Prof. Dr. Felipe Masiero Salvarani and Profa. Dra. Cíntia Daudt
On behalf of all co-authors
Reviewer 3 Report
Comments and Suggestions for Authors
I am generally positive for acceptance of the manuscript, but I would suggest to add the separate phylogenetic tree constructed based on the whole, no matter that the topology is the same
Author Response
Response to Reviewer 3
Comment:
I am generally positive for acceptance of the manuscript, but I would suggest to add the separate phylogenetic tree constructed based on the whole, no matter that the topology is the same.
Response:
We appreciate your positive evaluation and your valuable suggestion. As requested, we have now included a separate phylogenetic tree constructed using the whole-genome sequences (16AC18BR, 20AC18BR, and 34AC19BR) along with reference genomes from the PaVE database. Although the topology remains consistent with the partial L1-based tree, we agree that providing this additional analysis improves the completeness and transparency of our results. The new whole-genome phylogenetic tree has been added as Figure 6, and the corresponding description is included in the revised section:
“3.5. Whole-Genome Sequencing of Selected Isolates
Three samples (16AC18BR, 20AC18BR, and 34AC19BR) with high DNA quality were selected for whole-genome sequencing via high-throughput sequencing. The complete genomes ranged from 8,607 to 8,626 bp, and all included the canonical PV gene set: E1, E2, E4, E5, E6, E7, L1, L2, and a URR (upstream regulatory region). Genome coverage depth ranged from 20× to 35×, ensuring high confidence in base calling and genomic structure. All three genomes clustered within Lambdapapillomavirus 2 (CPV1), supporting the partial L1-based findings. These sequences are deposited at GenBank under accession numbers 16AC18_BR (PQ570013), 20AC18_BR (PQ570014), and 34AC18_BR (PQ570015). The corresponding phylogenetic tree, including these three genomes and reference sequences from the PaVE database, is presented in Figure 6."
